# Effects of Rehabilitation on Long-COVID-19 Patient’s Autonomy, Symptoms and Nutritional Observance

**DOI:** 10.3390/nu14153027

**Published:** 2022-07-23

**Authors:** Jeyniver Ghanem, Anne Passadori, François Severac, Alain Dieterlen, Bernard Geny, Emmanuel Andrès

**Affiliations:** 1Translational Medicine Federation of Strasbourg (FMTS), Team 3072 “Mitochondria, Oxidative Stress and Muscle Protection”, Faculty of Medicine, University of Strasbourg, 67000 Strasbourg, France; bernard.geny@chru-strasbourg.fr (B.G.); emmanuel.andres@chru-strasbourg.fr (E.A.); 2Institut IRIMAS (7499), Institut Universitaire de Technologie, Université de Haute-Alsace, 12 Rue des Frères Lumière, 68093 Mulhouse, France; alain.dieterlen@uha.fr; 3Centre de Réadaptation de Mulhouse—CRM, 57 Rue Albert Camus, 68200 Mulhouse, France; anne.passadori@arfp.asso.fr; 4Groupe Méthodes en Recherche Clinique, Service de Santé Publique, Hôpitaux Universitaires de Strasbourg, 67000 Strasbourg, France; francois.severac@chru-strasbourg.fr; 5Physiology and Functional Exploration Service, University Hospital of Strasbourg, 67091 Strasbourg, France; 6Service de Médecine Interne, Diabète et Maladies Métaboliques, Hôpitaux Universitaires de Strasbourg, 67000 Strasbourg, France

**Keywords:** COVID-19, rehabilitation, autonomy, malnutrition, long-haul symptoms

## Abstract

Background: Despite significant improvements in COVID-19 therapy, many patients still present with persistent symptoms and quality-of-life alterations. The aim of this study was to simultaneously investigate the long-term evolution of autonomy, malnutrition and long-lasting symptoms in people infected with COVID-19 and hospitalized in the ICU. Method: Patients’ clinical characteristics; extent of their loss of autonomy based on “Autonomie Gérontologie Groupes Iso-Ressources” (AG-GIR) classification; nutritional status while following the French and Global Leadership Initiative on Malnutrition (GLIM) recommendations; and symptom evolutions before infection, during hospitalization and rehabilitation, and up to 6 months after returning home were determined in thirty-seven patients. Results: Prior to a COVID-19 infection, all patients were autonomous, but upon admission to the rehabilitation center (CRM), 39% of them became highly dependent. After discharge from the center and 6 months after returning home, only 6 and 3%, respectively, still required considerable assistance. Of these thirty-seven patients, 11% were moderately malnourished and 81% presented with severe malnutrition, with a significant correlation being observed between malnutrition and autonomy (*p* < 0.05). Except for fatigue, which persisted in 70% of the patients 6 months after discharge from rehabilitation, all other symptoms decreased significantly. Conclusions: This study shows a striking decrease in autonomy associated with malnutrition after hospitalization for a COVID-19 infection and a clear beneficial effect from personalized rehabilitation. However, although almost all patients regained autonomy 6 months after returning home, they often still suffer from fatigue. Patient compliance with their nutritional recommendations deserves further improvement, preferably through personalized and persistent follow-up with the patient.

## 1. Introduction

As the SARS-CoV-2 pandemic has spread throughout the world, research has allowed us to increase our knowledge of this virus and to optimize COVID-19 infection prevention and treatment [1]. However, to date, studies concerning the nutritional status of patients after a COVID-19 hospitalization, in the long term, are relatively few and those concerning their autonomy are even fewer [2,3,4].

In fact, hospitalizations, whatever its causes, significantly affect both nutritional and autonomy statuses. These impacts have been observed previously, for example with influenza, in patients presenting with pneumonia [5,6,7,8]. Additionally, the SARS-CoV-2 disease initiates a hyper catabolic state [9,10], and studies showed a prevalence of malnutrition greater than 50% in severe coronavirus infections [11] and a reduced autonomy in people with severe infections [3]. Insufficient food intake to meet the sudden increase in the body’s need for anti-infective control during hospitalization and especially during ICU admission likely worsened the patient’s health. Furthermore, besides ICU admission, the length of hospital stays, the patient’s age and inflammation also favor impaired autonomy and nutritional statuses [11,12]. 

On the other hand, “long-haul” patients present with steady side effects, even in patients who have recovered from the SARS-CoV-2 disease. Thus, fatigue, cough, chest tightness, breathlessness, myalgia, loss of appetite, etc. are reported as “long COVID-19” symptoms. Studies showed that many patients who recovered and were discharged from hospitals showed persistence of at least one symptom at 60 days and that around 10% of them demonstrated symptoms even at 6 months after discharge [13,14,15].

Taken together, reduced autonomy, malnutrition and symptom persistence appear to be public health issues involving numerous patients presenting with significant impairment to their quality of life. Nevertheless, to date, despite its interest, to the best of our knowledge, no study has simultaneously analyzed the evolution of these three key parameters from before hospitalization until long after their return home.

The aim of this study was, therefore, to investigate the evolution of autonomy, malnutrition and long-lasting symptoms in people infected with COVID-19 during hospitalization in the ICU, after their rehabilitation and up to 6 months after returning home.

## 2. Methods

### 2.1. Population and Study Design

Between March and December 2020, at Mulhouse Rehabilitation Center, a lot of patients were hospitalized in the reanimation or intensive care units for SARS-CoV-2 infection. The study population consisted of thirty-seven patients aged 18 years or older who were admitted as inpatients to the Mulhouse Rehabilitation Center (CRM) and who then returned home. The participants gave their informed consent, and the study was approved by the Ethical Committee of the Strasbourg University (3 December 2020, CE-2020-197). Each patient, during his or her stay at the CRM, followed an individualized rehabilitation program based on the expertise of the health staff in order to optimize their recovery. The diverse, individualized programs for each patient do not allow us to present each of them here.

This non-interventional (with no control group) study analyzed each patient’s clinical and biological chart data obtained before their COVID-19 infection, during their hospitalization, and upon their admission and discharge from the CRM. Furthermore, via phone call with the patient 6 months after the end of their rehabilitation, a questionnaire was completed by a health staff member, thereby providing complete data.

The parameters collected from each patient included their demographic data (age, gender, medical history, etc.), their clinical variables (weight and height for calculation of the body mass index (BMI)), their independence status and their clinical symptoms of the SARS-CoV-2 infection.

This patient cohort is the result of the work by our clinical staff in structuring and consolidating the data obtained from the CRM patients who have met all of the inclusion criteria and who provided consent. The small size of this database was compensated by the quality of the data. 

### 2.2. Autonomy Determination

The extent of loss of autonomy was determined according to an “Autonomie Gérontologie Groupes Iso-Ressources” (AG-GIR) classification, allowing for an independence assessment according the national AGGIR grid [16,17]. There are six levels of GIR (GIR 1–6), GIR 1–2 correspond to the lowest level of autonomy, GIR 3 and 4 correspond to average levels of loss of autonomy and GIR 5–6 correspond to the highest level of autonomy.

### 2.3. Nutritional Status

The patients’ nutritional status was determined using their weight and body mass index (BMI), and variations were noted throughout the study. Abnormal nutrition was characterized by the presence of (i) the etiologic criteria—COVID-19 infection—and (ii) the phenotypic criteria defined by the French and Global Leadership Initiative on Malnutrition (GLIM) recommendations [18,19]. A loss > 5% was considered to be moderate malnutrition, and >10% was considered to be severe malnutrition. A questionnaire was completed 6 months after the patient’s return home in order to evaluate their food consumption.

### 2.4. Statistical Analysis

Continuous variables following a normal distribution are presented with mean +/− standard deviation (SD). The asymmetrical continuous variables are expressed as medians with interquartile ranges. The categorical variables are expressed as counts with percentages. The normality of the distribution was assessed graphically using histograms and the Shapiro–Wilk test. The evolution of autonomy (patients classified in GIR 5 and 6) was assessed using a mixed logistic regression model with a random “patient” effect to take into account repeated data over time. The evolution of the patient’s weight was assessed using a linear mixed model. The normality of the residuals and the random effects were checked graphically. The comparison between the length of stay and the lack of appetite was performed using the Wilcoxon rank-sum test. The statistical significance was set at *p* < 0.05. All of the data were anonymized and analyzed using the statistical software R version 4.1.1. R Core Team (2021). R: A language and environment for statistical computing. R Foundation for Statistical Computing, Vienna, Austria. URL https://www.R-project.org/ accessed on 1 January 2022.

## 3. Results

### 3.1. Flow Chart of the Study

The 37 patients were hospitalized between March and July 2020. The median length of the hospital stay was 41.0 [IQR: 20.0–60.0] days. In our population, 89.2% were admitted to the Intensive Care Unit (ICU) with a median length of stay of 23.5 [IQR: 13.0–35.0] days (Figure 1). Thereafter, the length of stay for rehabilitation at the CRM was 33.8 ± 31.9 days. Following this, all patients returned home.

### 3.2. Population Characteristics

As shown in Table 1, the population’s mean age was 64.3 ± 14.3 years. The average weight was 95.7 ± 23.2 kg, and the average BMI was 33 ± 6.6 kg/m^2^. Furthermore, 81.1% of the patients had a BMI superior to 25 kg/m^2^ before their illness. Men made up 68% of the patients. The most frequent comorbidities were diabetes (37.8%) and cancers (37.8%), followed by hypertension (35.1%) and cardiovascular disease (35.1%). The majority of the population did not smoke (64.9%) or were weaning off tobacco (32.4%).

### 3.3. Autonomy Evolution

The level of autonomy, as inferred from the GIR stages, is presented in Table 2. 

Prior to their infections, all patients (37) could autonomously perform their activities of daily living and all were categorized into the GIR 5 and 6 categories. Upon admission to the rehabilitation center, as much as 39% of patients were in GIR 1 and 2, which means that they became highly dependent. Despite the significant increase in the number of patients regaining autonomy and who were classified in GIR 5 and 6 between their admission to the CRM (30.5%), and their discharge from the CRM and return to normal life (69%), a high proportion of them still required assistance with most activities of daily living or required constant supervision (20% were in GIR < 5) (Figure 2). Furthermore, 6 months after their discharge from the CRM, only four patients were able to recover their independence and moved to categories GIR 5 and 6, whereas the others had more difficulty recovering their autonomy. 

Thus, considering that the highest levels of autonomy are GIR 5 and 6, after the initial striking decrease in autonomy levels due to the COVID-19 infection, the evolution was favorable since the number of autonomous patients grew from 11 (30.5%) to 30 (80%) 6 months after returning home. The evolution that began during rehabilitation continued slowly after the patients returned home. Nevertheless, as we can see, not all of the patients were able to restore their initial autonomy.

### 3.4. Evolution of Weight and BMI after COVID-19 Infection

Table 3 shows the changes in weight and BMI during the five phases of the study. Interestingly, both weight and BMI decreased from before hospitalization to admission to the rehabilitation center and slightly increased thereafter, likely in relation to the dietary monitoring and food supplements provided during rehabilitation. The weight and BMI decreases reached their lowest values of 77.6 ± 17.8 and 26.8 ± 5.2 kg/m^2^, respectively, at admission to the CRM compared with the initial value. 

Based on the recommendations for assessing malnutrition [18], we determined the percentages of weight loss for each individual from before pathology to admission to the rehabilitation center. Among the 37 patients, 11% were moderately malnourished (losses ≥ 5% in 1 month) and 81% presented with severe malnutrition (losses ≥ 10% in 1 month (Figure 3 and Figure 4). 

### 3.5. COVID-19 Symptom Evolution

The symptom evolution related to a COVID-19 infection is presented in Table 4. Interestingly, they varied depending on the different temporal periods studied: hospitalization, rehabilitation and 6 months after returning home.

Particularly, the fatigue identified subjectively in all of the individuals during their hospitalization persisted. Its prevalence at 6 months after the return home was as high as 70.2% (Figure 5). 

Cough, respiratory distress and dyspnea accompanied by fever, were present in almost all individuals during their hospitalization. Cough and fever were still present at the CRM but decreased sharply at 6 months after returning home. On the other hand, dyspnea was more persistent and even at 6 months after returning home, 13.5% of the population still complained of dyspnea and breathing difficulties.

Concerning gastrointestinal disorders, during hospitalization, 40.5% of individuals were subjected to more or less severe diarrhea and this discomfort strongly affected their food intake. In addition, 13.5% of individuals experienced nausea and vomiting. A lack of appetite was present in all of the individuals at the time of hospitalization and this persisted in 37.8% of people at the CRM. This disorder was often associated with asthenia, ageusia, nausea or vomiting. The patients with a lack of appetite had a median length of stay at the CRM of 48.0 [IQR: 22.5–61.5] days, in contrast with patients who had no lack of appetite and whose median length of stay at the CRM was of 20.0 [IQR: 9.5–28.0] days (*p* = 0.006).

The lack of appetite persisted in 18.9% of individuals 6 months after returning home.

Disorientation occurred during hospitalization in many patients (70.3%), but they later recovered. Aphasia was observed in 21.6% of patients during their hospitalization. Upon admission to the CRM, this disorder persisted in 13.5% of the patients. 

Interestingly, we observed that when patients presented with the lowest degree of autonomy (GIR 1–2), they often also presented with malnutrition and fatigue (Table 5).

### 3.6. Food Consumption Assessment 6 Months after Returning Home

To further investigate their health status and the causes of their malnutrition, we contacted patients 6 months after their return to normal life. According to the Canadian recommendations proposed by the Canadian Care Society and published on the Nutrition Care in Canada website [20], we established a post-COVID-19 dietary questionnaire to assess food consumption. The 36 responses obtained from the questionnaire are presented in Table 6. 

The consumption of fat (61.1%), meat (80.5%), fruit (50%) and vegetables (72.2%) was sufficient for the majority of patients, but the intake of dairy products was lower than recommended for 63.9% of them. Despite adequate starch consumption in 66.7% of the population, only four patients (11.1%) favored whole grains, bearing in mind that this recommendation was provided to all patients at the time of discharge from the CRM. Water intake was largely sufficient in this population (88.9%) and alcohol consumption was restricted, with a majority of 83.4% consuming no more than one drink per week.

## 4. Discussion

The main results of this study show (1) a striking decrease in autonomy associated with malnutrition after hospitalization for a COVID-19 infection, (2) a clear beneficial effect of personalized rehabilitation using these parameters and (3) that although almost all patients regained their autonomy 6 months after returning home, they often still suffered from fatigue.

Additionally, patient compliance with their diet recommendations deserves further improvement, preferably through a personalized and persistent follow-up with the patient.

### 4.1. Reduced Autonomy and Increased Malnutrition in Patients Hospitalized for COVID-19

The effects of COVID-19 on a patient’s autonomy deserves further investigations. In our study, all subjects were independent (GIR 5–6) prior to their infection but after their hospitalization and upon admission to the rehabilitation center, as much as 39% of the patients were highly dependent (GIR 1–2). Whether such a result was mainly due to COVID-19 per se or to the related hospitalization cannot be inferred from our data. However, a recent and interesting study showed that, compared with patients requiring a stay in the intensive care unit, people infected with COVID-19 were less likely to suffer from a loss of usual activities [3]. Thus, rather than COVID-19 alone, hospitalization appears to be an important cause of the loss of autonomy in patients infected with COVID-19. Accordingly, we observed that sarcopenia—defined as reduced muscle mass and strength—was related to the duration of hospitalization after COVID-19 infection [21], and a common hypothesis is that hospitalization and/or a SARS-CoV2-related increase in muscle weakness likely causes, in turn, a decrease in the GIR [22,23]. From this perspective, pre-existing comorbidities are also likely involved since they are associated not only with the need for hospitalization but also likely with persistent alteration [4,24].

Confirming previous studies [25], malnutrition as defined using the GLIM definition [18,19], was present in many of our patients and people registered in GIR 1 are more affected by the risk of malnutrition [26], and in a study investigating 4520 residents, residents who were malnourished showed an increase in overall dependency, including during mealtimes and while walking, which are major activities of daily living [27]. This reinforces the link between malnutrition and a patient’s functional status, including frailty and dependency [28,29,30]. 

### 4.2. Beneficial Effects of Rehabilitation on Patient’s Symptoms, Autonomy and Nutrition

As previously reported, many symptoms are associated with a COVID-19 infection, and they may concern all parts of the body. Our data confirm such an assumption, and it is of interest to note that almost all of the symptoms were reduced in the majority of patients after their rehabilitation. Thus, 6 months after returning home, respiratory, digestive and neurological alterations affected generally less than 10% of the patients, while they were present in more than 90% of them during hospitalization. Nevertheless, the percentage of patients complaining of fatigue decreased only from about 90 to 70%. This symptom is thus very common even late after a COVID-19 infection and is a characteristic of the post- or long-COVID-19 syndrome [31,32]. 

Fatigue might result from many mechanisms including mental and physical alterations [33,34], but mental mechanisms might play a role since after rehabilitation, upon discharge from the CRM, 69% of the patients were again independent and were categorized into the GIR 5–6 categories. This is much better than the 30.5% observed upon admission to the CRM but this also means that a significant number of patients did not completely recover their autonomy. Furthermore, 6% of the patients continued to experience severe loss in autonomy (GIR 1–2). This can be compared with the data demonstrating that 6 months after hospitalization, 4% of people infected with COVID-19 still presented with sarcopenia [21].

Concerning weight changes, the majority of our population was overweight before the COVID-19 pandemic started. This is consistent with previous data [4] showing a high rate of people with obesity being infected and placed in intensive care, and that might explain why patients did not reach their pre-hospitalization weight after rehabilitation. Indeed, nutritional recommendations systematically provided to the patients aimed to help them normalize their BMI. From this perspective, as shown by dairy products and whole grain consumptions, this progress should be useful. Indeed, only 5.5% of the patients appeared to know that they had received nutritional recommendations upon discharge from the CRM and to take them into consideration, 13.9% knew that they had received recommendations but never applied them, and the remaining 80.6% of patients were unaware that they had received any recommendations concerning nutrition. This poses a major problem and demonstrates the need for new methodologies for recommendations provided to patients who are discharged from institutions and returning to independent living. 

### 4.3. Limitations of the Study

The population sample is relatively small and lacks a comparison group (i.e., patients admitted to the ICU for other reasons or patients who did not receive rehabilitation). However, the patients’ data were well controlled and the literature allowed for a discussion with similar patients admitted to the ICU for other reasons. Furthermore, although investigated in detail, the symptoms reported by the patients could be subjective, which might be a limitation.

## 5. Conclusions

In summary, hospitalization for a severe SARS-CoV-2 infection demonstrated major adverse consequences on patients’ nutrition and autonomy, likely associated with many symptoms. 

Rehabilitation significantly improved such parameters, but this long-term study also shows that 6 months after returning home, 20% did not regain full autonomy and 70% of them still suffer from fatigue. Together with the fact that nutritional recommendations were only poorly followed, these data support a longer personalized follow-up with people hospitalized due to infections with COVID-19 long after the infection clears. A new strategy for recommendations and multifactorial follow-up with different healthcare experts—not only during rehabilitation but also after returning home—especially when malnutrition is established, are warranted.

## Figures and Tables

**Figure 1 nutrients-14-03027-f001:**
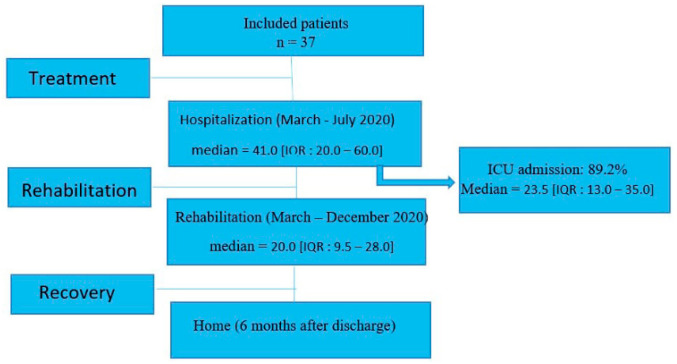
Flow chart of the study.

**Figure 2 nutrients-14-03027-f002:**
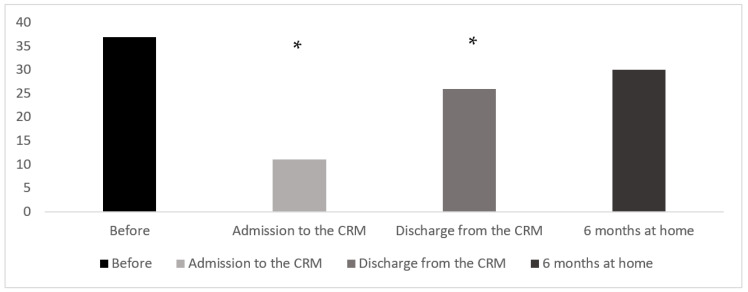
Autonomy recovery throughout the health care pathway, from before COVID-19 infection to 6 months after returning home (number of patients classified in GIR 5 and 6). * Sig diff between mobility score at admission to the rehabilitation center and at discharge.

**Figure 3 nutrients-14-03027-f003:**
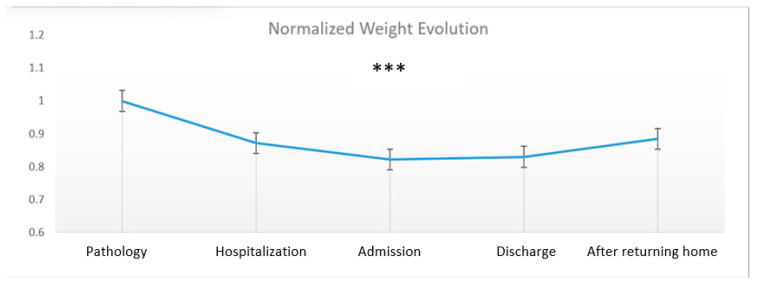
Normalized difference in weight at different temporal phases. *** *p* < 0.001 as compared to before the pathology.

**Figure 4 nutrients-14-03027-f004:**
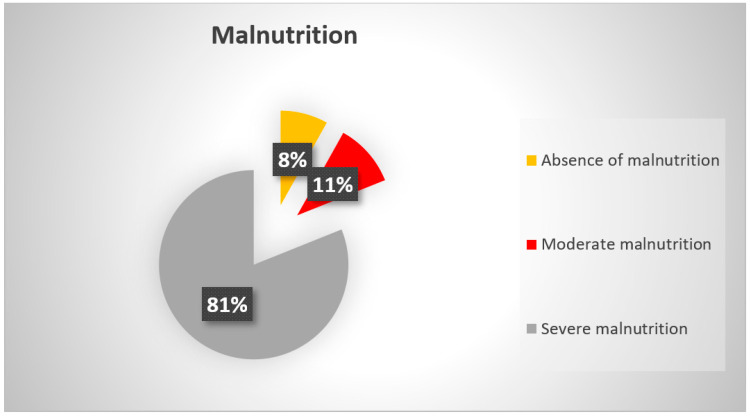
Degree of malnutrition due to COVID-19 at admission to the rehabilitation center.

**Figure 5 nutrients-14-03027-f005:**
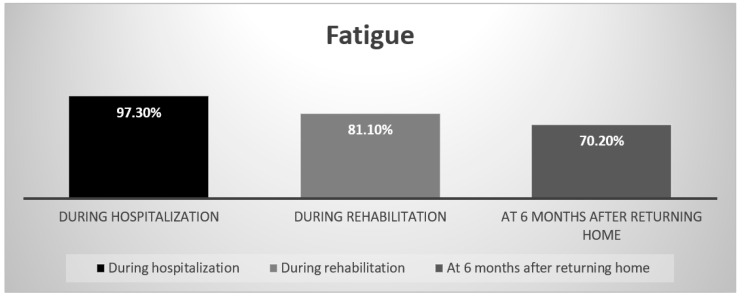
Fatigue persistence.

**Table 1 nutrients-14-03027-t001:** Population characteristics.

			*n* = 37	
Medical Characteristics			Average ± SD	
Age (years)			64.3 ± 14.3	
Weight (kg)			95.7 ± 23.2	
Height (m)			1.7 ± 0.1	
BMI (kg/m^2^)			32.9 ± 6.6	
			Effective	Percentage
Sex		Men	25	68%
		Women	12	32%
Diagnosis of COVID-19		PCR on nasopharyngeal swab	26	70.3%
		Compatible data scan	11	29.7%
Medical history			Effective	Percentage
Pre-illness weight status	Overweight		30	81.1%
	Normal weight		7	18.9%
Diabetes			14	37.8%
Cardiovascular diseases			13	35.1%
Stroke			6	16.2%
Dyslipidemia			7	18.9%
Hypertension			13	35.1%
Cancers			14	37.8%
COPD and chronic lung diseases			9	24.3%
Tobacco smoking	Smokers		1	2.7%
	Nonsmokers		24	64.9%
	Weaning off tobacco		12	32.4%
Renal failure			4	10.8%
Digestive disorders			6	16.2%
Psychiatric disorders			8	21.6%

**Table 2 nutrients-14-03027-t002:** Evolution of the patients’ autonomy from before COVID-19 infection to 6 months following their return home.

	GIR 5 & 6	GIR 3 & 4	GIR 1 & 2
GIR before COVID-19	100%	0	0
GIR at admission to the CRM	30.5%	30.5%	39%
GIR at discharge from the CRM	69%	25%	6%
GIR 6 months after returning home	80%	17%	3%

CRM: centre de readaptation de mulhouse=mulhouse rehabilitation center; GIR; groupe iso-ressources.

**Table 3 nutrients-14-03027-t003:** Weight and BMI variations during the patient’s follow-up.

Average ± SD	Before the Pathology	During Hospitalization	At Admission to the CRM	At Discharge from the CRM	At 6 Months after Returning Home
Weight (kg)	95.8 ± 23.2	83 ± 18.9	77.6 ± 17.8	78.8 ± 18.3	83.7 ± 17.6
BMI (kg/m^2^)	32.9 ± 6.6	28.4 ± 5.7	26.8 ± 5.2	27.2 ± 5.5	28.9 ± 5.1

CRM: Mulhouse Rehabilitation Center.

**Table 4 nutrients-14-03027-t004:** Evolution of COVID-19 symptoms.

		During Hospitalization	During Rehabilitation	At 6 Months after Returning Home
Fatigue		97.3%	81.1%	70.2%
Respiratory disorders	Dyspnea	97.3%	75.7%	13.5%
	Cough	86.5%	35.1%	2.7%
Digestive disorders	Lack of appetite	97.3%	37.8%	18.9%
	Diarrheas	40.5%	37.8%	2.7%
	Nausea and vomiting	13.5%	13.5%	0%
Fever		91.9%	18.9%	0%
Neurological disorders	Disorientation	70.3%	21.6%	2.7%
	Ageusia	32.4%	32.4%	8.2%
	Anosmia	29.7%	27%	2.7%
	Aphasia	21.6%	13.5%	2.7%

**Table 5 nutrients-14-03027-t005:** Relationship between malnutrition, autonomy and fatigue.

		Malnutrition at Admission			Autonomy at Admission			Fatigue at Admission
		Absent	Moderate	Severe	GIR 5 & 6	GIR 3 & 4	GIR 1 & 2	
Malnutrition at admission	Absent	3	0	0	1	2	0	2
	Moderate	0	4	0	2	1	1	3
	Severe	0	0	30	8	8	14	25
Autonomy at admission	GIR 5 & 6	1	2	8	11	0	0	6
	GIR 3 & 4	2	1	8	0	11	0	11
	GIR 1 & 2	0	1	14	0	0	15	13
Fatigue at admission		2	3	25	6	11	13	30

**Table 6 nutrients-14-03027-t006:** Recommendations provided during post-COVID-19 rehabilitation compared to dietary habits at 6 months after discharge.

Recommendations Applied to CRM	Question	Categories	Number	Percentage
Small frequent meals (3–5 meals a day)	Number of meals/days	Between 3 and 5 meals	30	83.4%
		<3 meals	3	8.3%
		>5 meals	3	8.3%
Consume protein foods at every meal (>6 time)	Meat consumption/week	>6 times	29	80.5%
		Between 3 and 6	6	16.7%
		3 times	1	2.8%
Consume dairy products at each meal (between 3 and 4 servings)	Dairy products consumption/day	Between 3 and 4 servings	8	22.3%
		<2 servings	23	63.9%
		≥5 servings	5	13.8%
Consume fruits (between 2 and 4 servings)	Fruit consumption/day	Between 2 and 4 servings	18	50%
		≥5 servings	10	27.8%
		≤1 serving	8	22.2%
Consume vegetables at each meal	Do you consume vegetables at each meal?	Yes	26	72.2%
		No	10	27.8%
Consume fat at each meal	Do you consume fat at each meal?	Yes	22	61.1%
		No	14	38.9%
Consume starchy foods at each meal (between 3 and 4 servings)	Starch consumption/day	Between 3 and 4 servings	24	66.7%
		≥5 servings	9	25%
		<2 servings	3	8.3%
Choose whole grains	Do you prefer whole grains?	Yes	4	11.1%
		No	32	88.9%
Drink 8–10 cups of liquid per day	Fluid intake/day	Between 8 and 10 cups	2	5.55%
		>10 cups	32	88.9%
		≤7 cups	2	5.55%
---	Alcohol consumption/week	≤1 cup	30	83.4%
		Between 1 and 2 cups	5	13.8%
		>2 cups	1	2.8%

## Data Availability

Not applicable.

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
