# Peer review of "Effects of Rehabilitation on Long-COVID-19 Patient’s Autonomy, Symptoms and Nutritional Observance"

_nutrients, 2022, doi:10.3390/nu14153027_

Round 1

Reviewer 1 Report

Dear authors,

In this paper you describe results of an interesting research regarding covid sequelae. I suggest to provide minor language revision and change figure 1 with a more clear flowchart

Author Response

Dear Reviewer, 1,

Thank you for your nice comments concerning our paper. Please find the specific responses to your query, the document had language revision.

Particularly, the new flow chart is clearer allowing describe population follow-up and clinical evolution.

Reviewer 2 Report

Analyzing the long-lasting effects of severe Covid-19 is a topic of particular interest. Specifically, nutritional status and autonomy for the activities of daily living could be profoundly affected.  The special circumstances lived during the pandemic made it difficult to design appropriate studies. However, this study has certain flaws that could be avoided.

Introduction

The authors should include information about the reasons for losing autonomy and worsening nutritional status after UCI admission by COVID-19.

"Around many patients" is not a very precise statement

References about nutritional and functional status should be cited independently (Lines 48-50).

Methods

The sample is small and lacks a comparison group (i.e patients admitted to ICU for other reasons, patients who did not receive rehabilitation). For these reasons, it seems difficult to extract clear conclusions. These issues should be recognized as limitations of the study.

The manuscript lacks information on the registration of the data. The authors should specify how was each data registered. Did patients report the data or did the clinical staff record the data? Data reported by the patients could be subjective and therefore this also should be recognized as a limitation.

There is no information about the rehabilitation process received by the patients. Were they outpatients or inpatients? The authors should also specify the characteristics of the rehabilitation program such as the type and intensity of the exercises, the frequency, and the duration of the sessions, and data about nutritional supplementation.  This information is absolutely necessary for understanding the recovery of nutritional and autonomy status in these patients.

There is a clear discordance between the statistical analysis described in the methods and that presented in the results.

How was normality assessed?

The authors should specify which variables are compared with each test. 

According to the reading of the manuscript,  ANOVA and student´s t-test seem to compare different types of variables.  However, both, analyze the effects of categorical variables on a quantitative variable. Authors should specify which comparisons are performed by paired and unpaired measures

The authors presented some correlations in the manuscript. Nevertheless, they do not appear in the description of the statistics.

Considering the number of points assessed over time, repeated-measures ANOVA, followed by a posthoc test seems to be more appropriate than the paired t-test. 

Results

It does not seem clear the meaning of the asterisks in Table 2 and Figure 2. Which group are they comparing?

The meaning of CRM should be specified in the legend of Table 3.

In the result section, the authors defined moderate and severe malnutrition based on (weight?) losses of > 5% or > 10% respectively. This definition should be included in the methods section.

The authors reported a correlation between the prevalence of malnutrition and the GIR value. The prevalence of malnutrition is a characteristic of a population. However, this correlation seems to have been applied to individuals. Are the authors correlating the severity of malnutrition (weight loss) and the GIR value?

In Figure 3, the authors should compare the differences between different points, not only between the first and the last points. 

In the legend of  Figure 3, the authors should specify when was recorded malnutrition showed in the Figure.

How were recorded symptoms shown in Table 5 and Figure 6? Are they comparing data recorded by the clinical staff and reported by the patients? Particularly, how was fatigue assessed? Did they use any validated scale?

This table compares the results of two periods (hospitalization and rehabilitation) and the results of a time point (6 months after finishing rehabilitation). The length of the periods seems different. In consequence, any symptom could be more prevalent in longer periods? 

P values should be represented consistently over the manuscript (exact p value or p<)

Which was the statistical test used to compare results in Table 5? 

The tool to assess food consumption presented in Table 6 was not explained in the Methods section

Discussion

The authors stated that rehabilitation was personalized. In the Methods section, this should be explained.

Section 4.1 appears twice

Line 254: "Concerning weight evolution, the majority of our population demonstrated a severe increase in BMI before COVID occurrence".  Authors did not measure that there was an increase in BMI before COVID occurrence

This work has certain limitations that should be recognized by the authors at the end of the discussion (design, sample, recording of the results, and so on)

COVID-19 should be spelled consistently

Author Response

Dear Reviewer, 2,

Thank you for your nice comments concerning our paper. Please find below the specific responses to our queries, we modified accordingly the manuscript as shown below.

Introduction

The authors should include information about the reasons for losing autonomy and worsening nutritional status after UCI admission by COVID-19.

« An insufficient food intake to meet the sudden increase in the body's need for anti-infective control during hospitalization and more especially during ICU admission likely worsened the health state. Further, besides ICU admission, length of hospital stays, patients age and inflammation also favor impaired autonomy and nutritional status (11,12).».

"Around many patients" is not a very precise statement

Ok, Modified.

“Studies showed that many patients recovered and discharged…”

References about nutritional and functional status should be cited independently (Lines 48-50).

Ok, we previously added a new ref 12. Then, references concern long haulers symptoms.

Methods

The sample is small and lacks a comparison group (i.e., patients admitted to ICU for other reasons, patients who did not receive rehabilitation). For these reasons, it seems difficult to extract clear conclusions. These issues should be recognized as limitations of the study.

Ok, thanks. We state this in a new limitation paragraph.

Limitations of the study.

The population sample is relatively small and lacks a comparison group (i.e., patients admitted to ICU for other reasons, patients who did not receive rehabilitation). However, patient’s data were well controlled, and literature allowed discussion with similar patients admitted to ICU for other reasons.

Ok, thanks. We state this in a new limitation paragraph.

The manuscript lacks information on the registration of the data. The authors should specify how was each data registered. Did patients report the data or did the clinical staff record the data? Data reported by the patients could be subjective and therefore this also should be recognized as a limitation.

“This patient cohort is the result of the work of our clinical staff in structuring and consolidating the data of the CRM patients who have met all the inclusion criteria up to consent. “

However, indeed, data reported by the patients could be subjective and therefore this is now recognized as a limitation.

Limitations of the study.

Further, although investigated in detail, the symptoms reported by the patients could be subjective that might be a limitation.

There is no information about the rehabilitation process received by the patients. Were they outpatients or inpatients? The authors should also specify the characteristics of the rehabilitation program such as the type and intensity of the exercises, the frequency, and the duration of the sessions, and data about nutritional supplementation.  This information is absolutely necessary for understanding the recovery of nutritional and autonomy status in these patients.

“Each patient during his or her stay at the CRM follows an individualized rehabilitation program based on the expertise of the health staff to optimize they recovery. The diversity due to the individualization of programs does not allow us to represent each of them.”

There is a clear discordance between the statistical analysis described in the methods and that presented in the results. How was normality assessed? The authors should specify which variables are compared with each test. 

Continuous variables following a normal distribution are presented with mean +/- standard deviation (SD). Asymmetric continuous variables are expressed as medians with interquartile ranges. Categorical variables are expressed as counts with percentages. The normality of the distribution was assessed graphically using histograms and with the Shapiro-Wilk test. The autonomy evolution (patients classified in GIR 5 and 6) was assessed using mixed logistic regression model with a random « patients » effect to take into account the repeated data over time. The evolution of weight was assessed using linear mixed model. The normality of residuals and random effects was checked graphically. The comparison of the length of stay according to the lack of appetite was performed using the Wilcoxon rank-sum test. Statistical significance was set at p<0.05. All data were anonymized and analyzed using the statistical software R version 4.1.1. R Core Team (2021). R: A language and environment for statistical computing. R Foundation for Statistical Computing, Vienna, Austria. URL https://www.R-project.org/.

According to the reading of the manuscript, ANOVA and student´s t-test seem to compare different types of variables.  However, both, analyze the effects of categorical variables on a quantitative variable. Authors should specify which comparisons are performed by paired and unpaired measures. Considering the number of points assessed over time, repeated-measures ANOVA, followed by a posthoc test seems to be more appropriate than the paired t-test. 

Patients with a lack of appetite had a median length of stay at the CRM of 48.0 [IQR: 22.5 – 61.5] days, in contrast with patients without a lack of appetite who had a median length of stay at the CRM of 20.0 [IQR: 9.5 – 28.0] days (p=0.006).

The 37 patients were hospitalized between March and July 2020. The median length of hospital stay was 41.0 [IQR: 20.0 – 60.0]. In our population, 89.2% were admitted in Intensive Care Unit - ICU with a median length stay of 23.5 [IQR: 13.0 – 35.0] (Figure 1). Thereafter, the length stay for rehabilitation at the CRM was 33.8 ± 31.9 days.

Results

It does not seem clear the meaning of the asterisks in Table 2 and Figure 2. Which group are they comparing?

Added in the legend of figure2

*Sig diff between mobility score at the admission to the rehabilitation center and at the discharge

The meaning of CRM should be specified in the legend of Table 3.

Yes, done.

In the result section, the authors defined moderate and severe malnutrition based on (weight?) losses of > 5% or > 10% respectively. This definition should be included in the methods section.

Ok, done.

“A loss > 5 % was considered as moderate malnutrition and >10% was considered as severe malnutrition.

The authors reported a correlation between the prevalence of malnutrition and the GIR value. The prevalence of malnutrition is a characteristic of a population. However, this correlation seems to have been applied to individuals. Are the authors correlating the severity of malnutrition (weight loss) and the GIR value?

Yes, we modified accordingly the text.

In Figure 3, the authors should compare the differences between different points, not only between the first and the last points. 

The problem is that during hospitalization we don’t have a specific time for weight registration and at home weight was taken by the patient itself. For that we chose these two points that are to most accurate to be compared specially to evaluate the effect of the COVID-19 pathology before the beginning of the rehabilitation

In the legend of Figure 3, the authors should specify when was recorded malnutrition showed in the Figure.

Modified in the legend of Figure 3

How were recorded symptoms shown in Table 5 and Figure 6? Are they comparing data recorded by the clinical staff and reported by the patients? Particularly, how was fatigue assessed? Did they use any validated scale?

Yes, they were recorded by the clinical staff, as reported by the patient. Fatigue was a subjective question to the patient: do you feel fatigue, did you feel your energy back, the answer was yes or no. We acknowledge such limitation both in the text adding “subjectively” and in the limitation paragraph.

« Particularly, the fatigue identified subjectively in all the individuals…”

This table compares the results of two periods (hospitalization and rehabilitation) and the results of a time point (6 months after finishing rehabilitation). The length of the periods seems different. In consequence, any symptom could be more prevalent in longer periods? 

Indeed, length of hospitalization and rehabilitation periods can differ between patients, but the 6-month time point after rehabilitation is the same for all patients. Thus, other evolution at longer time points is not investigated by our study. However, 6 months after returning home appears a good point since it allows to determine a relatively stable condition of the patients.

P values should be represented consistently over the manuscript (exact p value or p<)

Ok. Resolved all over the article

Which was the statistical test used to compare results in Table 5? 

No statistical result in table 5

The tool to assess food consumption presented in Table 6 was not explained in the Methods section

ok, completed. \

“A questionnaire was used 6 months after returning home to evaluate food consumption”

Discussion

The authors stated that rehabilitation was personalized. In the Methods section, this should be explained.

“Each patient during his or her stay at the CRM follows an individualized rehabilitation program based on the expertise of the health staff to optimize they recovery. The diversity due to the individualization of programs does not allow us to represent each of them.”

Section 4.1 appears twice

Ok thanks. We modified.

Line 254: "Concerning weight evolution, the majority of our population demonstrated a severe increase in BMI before COVID occurrence".  Authors did not measure that there was an increase in BMI before COVID occurrence

Explain differently and more clearly.

Ok thanks. “Concerning weight evolution, the majority of our population was overweight before COVID-19 occurrence.”

This work has certain limitations that should be recognized by the authors at the end of the discussion (design, sample, recording of the results, and so on). Analyzing the long-lasting effects of severe Covid-19 is a topic of particular interest. Specifically, nutritional status and autonomy for the activities of daily living could be profoundly affected.  The special circumstances lived during the pandemic made it difficult to design appropriate studies. However, this study has certain flaws that could be avoided. *

Yes, thanks, we acknowledge this and added a limitation paragraph.

COVID-19 should be spelled consistently

Resolved

Round 2

Reviewer 2 Report

Authors took into account my comments